# Molecular Markers for Marker-Assisted Breeding for Biotic and Abiotic Stress in Melon (*Cucumis melo* L.): A Review

**DOI:** 10.3390/ijms25126307

**Published:** 2024-06-07

**Authors:** Durre Shahwar, Zeba Khan, Younghoon Park

**Affiliations:** 1Plant Genomics and Molecular Breeding Laboratory, Department of Horticultural Bioscience, Pusan National University, Miryang 50463, Republic of Korea; dsh92@pusan.ac.kr; 2Center for Agricultural Education, Faculty of Agricultural Sciences, Aligarh Muslim University, Aligarh 202002, India; khanzeba02@gmail.com; 3Life and Industry Convergence Research Institute, Pusan National University, Miryang 50463, Republic of Korea

**Keywords:** quantitative trait loci, genetic and molecular analyses, melon (*Cucumis melo* L.), biotic stress, abiotic stress

## Abstract

Melon (*Cucumis melo* L.) is a globally grown crop renowned for its juice and flavor. Despite growth in production, the melon industry faces several challenges owing to a wide range of biotic and abiotic stresses throughout the growth and development of melon. The aim of the review article is to consolidate current knowledge on the genetic mechanism of both biotic and abiotic stress in melon, facilitating the development of robust, disease-resistant melon varieties. A comprehensive literature review was performed, focusing on recent genetic and molecular advancements related to biotic and abiotic stress responses in melons. The review emphasizes the identification and analysis of quantitative trait loci (QTLs), functional genes, and molecular markers in two sections. The initial section provides a comprehensive summary of the QTLs and major and minor functional genes, and the establishment of molecular markers associated with biotic (viral, bacterial, and fungal pathogens, and nematodes) and abiotic stress (cold/chilling, drought, salt, and toxic compounds). The latter section briefly outlines the molecular markers employed to facilitate marker-assisted backcrossing (MABC) and identify cultivars resistant to biotic and abiotic stressors, emphasizing their relevance in strategic marker-assisted melon breeding. These insights could guide the incorporation of specific traits, culminating in developing novel varieties, equipped to withstand diseases and environmental stresses by targeted breeding, that meet both consumer preferences and the needs of melon breeders.

## 1. Introduction

Melons *(Cucumis melo* L.; 2n = 2x = 24) grow in warm areas worldwide and belong to the family Cucurbitaceae, including cucumbers, pumpkins, and squashes. Muskmelon is indigenous to tropical Africa, specifically in the eastern region below the Saharan Desert [1]. The exact origin of melon domestication is debated; some suggest India due to ancient cultivation, while others lean towards Iran. However, the dominant belief is that melons, similar to related species, originated in Africa. Historical evidence shows their early presence in Egypt and Iran, which later spread across Asia, becoming significant in India and China. The melon family includes various varieties with different flavors, textures, and appearances, of which cantaloupe, honeydew, Persian, and watermelon are common varieties. Each variety has optimal growth conditions, leading to diverse global production landscapes. Melon shows high genetic variability, visible in various dimensions such as fruit size, flesh color, and sex determination (i.e., monoecious and andromonoecious) [2], thus exhibiting a huge polymorphism. Melons are annual trailing vines with broad leaves and yellow flowers. The melon fruit exhibits diversity in shape, size, and color, which vary according to the specific variety. This diversity includes well-known cantaloupes with netted rinds and smooth-skinned varieties, such as honeydew. Melons are not only enjoyed fresh, but also find applications in various culinary dishes, beverages, and desserts. The fruit flesh is a rich source of essential nutrients, including calcium, magnesium, potassium, iron, and zinc. It also contains sugars such as glucose, sucrose, and fructose, along with organic acids, particularly myristic and pantothenic acids. Furthermore, melons provide amino acids and a spectrum of phytochemicals, including carotenoids such as alpha- and beta-carotene. Essential vitamins, such as A, C, E, thiamin, riboflavin, and niacin, are present in addition to dietary fiber and antioxidant enzymes [3]. *Cucumis melo* has been studied for its beneficial medicinal properties, which include antiulcer, analgesic, anti-inflammatory, free radical scavenging, antioxidant, anthelmintic, antimicrobial, hepatoprotective, antidiabetic, anticancer, and antifertility activities [4].

According to FAO data, the global melon output is expected to exceed 40 million tons per year, with China being the primary producer (12.7 million tons per year) [5]. The seed market for cultivated melons is significant, with various seed companies offering a range of melon cultivars. The international market for melon-derived beverages, including juices and nectar, is expected to exceed USD 3.5 billion by 2020. The projections suggest a compound annual growth rate of more than 6% from 2021 to 2028 [6]. Despite growth in production, the melon industry faces several challenges. Plants are exposed to a wide range of biotic and abiotic stresses throughout their growth and development; consequently, their final yield is negatively affected by numerous diseases, pests, and abiotic stresses [7]. The primary biotic and abiotic stresses that affect plants are geographic location, climatic conditions, and the melon cultivar. Abiotic stressors, including excessive temperature, high salt conditions, drought, and flooding, diminish crop yields, leading to an average yield reduction in most crops of approximately 50% [8]. Genetic studies in melons have been performed to identify functional genes that confer resistance to various diseases and abiotic factors to aid in the breeding of resistant varieties and provide more genetic information on melon species. Several genes that can be transcriptionally modified in response to stress (biotic and abiotic) have been identified and functionally characterized through the development of various omics technologies, suggesting that they participate in the maintenance of stress tolerance [9]. The important quantitative trait loci (QTLs) identified in previous studies are briefly discussed in Table 1.

Most cucurbit breeding endeavors prioritize the identification of disease resistance and associated molecular markers. Melon crop production is usually impeded by a wide range of biotic and abiotic stresses that negatively affect crop quality and yield [24]. Multiple stressors have a substantial impact on melon production in most parts of the world, particularly in Asia and Africa. Drought, salinity, high temperatures, and cold negatively affect plant growth and development. Traditionally, pesticides, fungicides, and insecticides have been used for disease management. Nonetheless, their long-term use has been discouraged owing to the negative environmental repercussions of extended applications [25]. Because of the increasing reliance on pesticides and insecticides, there has been a paradigm shift toward a greater emphasis on genetic disease resistance in melon plants. This shift in emphasis has emerged as a key goal of current breeding research efforts [11].

The development of melon varieties that are resistant to pests, diseases, and abiotic stresses is essential to ensure the long-term sustainability of melon production. Breeders can develop melon varieties that are productive and resilient to climate change challenges using different breeding approaches. New tools utilizing DNA markers for marker-assisted breeding can be implemented to achieve a gradual increase in yield and boost production. Therefore, it is imperative to utilize diverse high-yield lines as foundational materials for incorporating beneficial attributes or genes from otherwise unadapted cultivars, lines, or wild relatives. The prevalent perception is that indirect marker-assisted selection (MAS) can considerably improve the efficiency and efficacy of conventional breeding [26]. The emergence of novel biotypes and environmental stresses has necessitated the integration of multiple resistance genes into high-yield cultivars to endow them with an expanded spectrum of resistance. This improved resistance allows these cultivars to tolerate concurrent disease attacks and thrive under harsh climatic conditions.

Recently, several hybrids have been developed to improve disease resistance and resilience to abiotic stressors. Many economically important characteristics of melons, such as yield, qualitative traits, and disease resistance, have a polygenic mode of inheritance. Owing to their complicated characteristics and vulnerability to environmental interactions, identifying QTLs using traditional breeding approaches is challenging. Nevertheless, advances in DNA marker techniques and linkage mapping have made it possible to unravel the complexities inherent in quantitatively inherited features [27]. MAS allows for the indirect identification of specific genetic characteristics in offspring produced by the controlled mating of compatible parental genotypes. Multiple factors, including the complexity of the desired trait and proximity of the genetic marker to the target characteristic influence the efficiency of MAS. MAS is becoming increasingly important for improving plant breeding efficiency by enabling the precise transfer of genomic regions of interest and accelerating the recovery of genome segments bearing the desired characteristics [28].

This review delves into the genetic mechanisms governing melons’ responses to both biotic and abiotic stress factors, with the objective of guiding trait selection and the development of resilient melon varieties capable of withstanding diseases and adverse environmental conditions. The remainder of this paper is bifurcated into two sections. The initial part explores QTLs, pivotal and supplementary functional genes, and the establishment of molecular markers associated with biotic and abiotic traits. These encompass viral, bacterial, and fungal pathogens, as well as nematodes, cold/chilling, drought, salt, and toxic compounds. The subsequent section briefly outlines the molecular markers utilized for marker-assisted backcrossing (MABC) and the identification of cultivars resistant to both biotic and abiotic stressors. The emphasis lies in highlighting their significance in strategic marker-assisted melon breeding, facilitating the integration of desirable traits into new varieties.

## 2. Methodology

The relevant peer-reviewed publications for this study were obtained from the following databases:Science Direct (www.sciencedirect.com, accessed on 23 June 2023)Research gate (www.researchgate.net, accessed on 28 June 2023)Google Scholar (scholar.google.com, accessed on 20 July 2023)Web of Science TM (www.webofknowledge.com, accessed on 2 August 2023)Scopus (www.scopus.com, accessed on 26 August 2023)

The primary search terms used were: “*Cucumis melo*”, “biotic stress”, “abiotic stress”, “genetic analysis”, “QTL”, “gene mapping”, “molecular markers” “candidate gene identification”, and “marker assisted selection”. Secondary or additional search terms for biotic stress included: “genetic study of viral pathogen in melon”, “genetic study of fungal pathogens in melon”, “genetic analysis of bacterial pathogens in melon”, genetic study of nematodes in melon”, genetic analysis of bacterial pathogens in melon”, genetic study of insect and herbivore in melon”, “genetic analysis of cold/chilling stress in melon”, “genetic study of drought stress in melon”, “genetic analysis of salt stress in melon”, and “genetic analysis of toxic compounds stress in melon”. The tertiary search terms used were: “identification of candidate gene for viral pathogens in melon”, “identification of candidate gene for fungal pathogens in melon”, “identification of candidate gene for bacterial pathogens in melon”, “identification of candidate gene for nematodes in melon”, “identification of candidate gene for insect and herbivore in melon” “identification of candidate gene for cold/chilling stress in melon”, “identification of candidate gene for drought stress in melon”, “identification of candidate gene for salt stress in melon”, and “identification of candidate gene for toxic compounds in melon”.

For biotic stress, further detailed and expanded search terms were used: “identification of candidate gene for cucumber mosaic virus disease in melon”, “identification of candidate gene for cucumber green mottle mosaic virus disease in melon”, “identification of candidate gene for tomato leaf curl New Delhi virus in melon”, “identification of candidate gene for zucchini yellow mosaic virus in melon”, “identification of candidate gene for Fusarium wilt disease in melon”, “identification of candidate gene for gummy stem blight disease in melon”, “identification of candidate gene for powdery mildew disease in melon”, “identification of candidate gene for bacterial fruit blotch disease in melon”, and “identification of candidate gene for Root-knot nematodes in melon”. To retrieve studies on marker-assisted backcrossing and molecular markers, we used the search term “molecular markers for marker-assisted backcrossing (MABC) and cultivar identification”. The most relevant publications, irrespective of their publication year, were included in this study.

## 3. Molecular Markers for Biotic and Abiotic Stress in Melons

Throughout their growth and development, melons encounter diverse biotic and abiotic stressors that pose significant threats to their vitality. In the contemporary context of anthropogenic global warming and ensuing climatic change, aridity is rising in traditional production areas, placing melon cultures under more abiotic stress and reducing yield. Different abiotic stresses, such as salt, osmotic stress, and heavy metals, are experienced by plants growing in different environments and soil conditions. However, plants can resist these stresses to some extent owing to their acclimatization processes, such as detoxification. In particular, an increase in drought stress has been observed, whereas salt stress is a direct result of water scarcity because excessive irrigation increases salt accumulation in the soil. Plant responses to stress are significantly correlated with changes in protein accumulation, resulting in the reprogramming of gene expression and phenotypic modifications that affect stress tolerance [29].

The adaptability of plant development becomes a dynamic objective, influenced by climate change, encompassing fluctuations in temperature, precipitation, and air composition, among other factors. Simultaneous environmental shifts may foster the onset of new plant infections and pests or elevate the prevalence of existing ones. The combined effects of abiotic and biotic stresses are anticipated to have an adverse impact on plants’ molecular functions, developmental processes, morphological features, and physiology, leading to a significant reduction in crop output and quality as a result of this complex environmental scenario. Enhancements in muskmelon crops, concerning yield, quality, and resistance to diseases and pests, have primarily been achieved through conventional breeding methods. Principal and secondary diversity centers house a substantial number of resistant melon accessions, making them invaluable sources of germplasm for melon breeders. Therefore, the use of these accessions is crucial for improving melon resistance, and has a significant impact on varietal development.

### 3.1. Biotic Stress

The enhancement of muskmelons, focusing on yield, quality, and resistance to diseases and pests, has predominantly been achieved through conventional breeding methods. Primary and secondary diversity centers are home to the majority of resistant melon accessions, rendering them valuable germplasm sources for melon breeders. Therefore, using these accessions is crucial for improving melon resistance, and has a significant impact on varietal development. Plant developmental adaptability is a moving objective owing to climate change, which includes variations in temperature, precipitation, and air composition, among other factors. Parallel environmental changes may encourage the emergence of new plant infections and/or pests, or increase the prevalence of existing ones.

#### 3.1.1. Viral Pathogens

The cucumber mosaic virus (CMV) is a prevalent viral pathogen known to affect cucurbits. Over 80 aphid species have been identified as facilitators of viral transmission, employing a non-persistent, stylet-borne mechanism [30]. Susceptibility to CMV varies among different cucurbit hosts, with watermelons being the least severely affected, whereas squashes, cucumbers, and pumpkins are more severely affected. Foliar symptoms include the bending of petioles, curling of leaves, and yellow spotting, which, in advanced stages, culminate in the formation of mosaicism and extensive curling of the leaf. Fruits exhibit surface irregularities and occasional color variations, which reduce their quality and marketability [31].

The development of resistance to CMV through breeding poses challenges because of the numerous strains that can overcome resistance genes effective against other strains [32]. To date, no commercially available melon cultivars with resistance to CMV, tailored for western production regions, have been identified. Essafi et al. [16] identified a QTL linked to CMV resistance on linkage group XII of the exotic melon variant PI 161375, known as “Sonwang Charmi” (SC). They used near-isogenic lines, which were blends of SC genes against a backdrop of the CMV-susceptible variant Piel de Sapo (PS). Further exploration pinpointed the resistance locus, termed *cmv1*, situated in a span of 2.2 cM, bounded by markers *CMN61_44* and *CMN21_55*. However, efforts to associate this resistance with any of the ten translation initiation factors in the melon genome did not yield any positive results within the defined genetic interval. This led to the hypothesis that the CMV resistance of SCs may stem from multiple genes reacting to varying CMV strains. One of these genes, *cmv1*, provides complete protection against the virus [16]. Pascual et al. [33] highlighted that the mechanism of resistance against CMV in melons is typically recessive and often relies on multiple genes that exhibit quantitative properties. However, they emphasized that resistance against specific CMV strains depends on a previously identified single gene, *cmv1*. This gene acts as a barrier, confining the virus to bundled sheath cells, thereby averting full-blown infection. This containment action is influenced by a viral movement protein (MP). *cmv1* encodes vacuolar protein sorting 41 (*CmVPS41*), a protein associated with the transport of proteins from the Golgi apparatus to the vacuole via late endosomes [33]. Furthermore, Pascual et al. [33] examined variations in *CmVPS41* across 52 melon variants spanning 15 distinct melon groups from both the spp. *melo* and spp. *agrestis* categories. They recognized 16 unique gene sequences, which translated into 12 different CmVPS41 protein forms. After testing all gene sequences, they identified nine novel resistance types that appeared to be linked to two specific mutations, L348R or G85E. Interestingly, in these newly identified resistant variants, although the virus could still multiply and transfer between cells, it could not infiltrate the phloem. This suggests that blocking the virus from accessing the phloem could be a widespread resistance strategy for melons steered by CmVPS41. Therefore, the principal function of CmVPS41 in melons is to act as a wide gatekeeper promoting resistance to CMV phloem invasion [33].

Cucumber green mottle mosaic virus (CGMMV) poses a severe risk to the cultivation of melons, watermelons, and cucumbers. CGMMV is a member of the *Tobamovirus* genus, family Virgaviridae [34], and affects cucurbitaceous plant leaves by causing systemic mottling and mosaic-like symptoms. Young melon leaves exhibit initial signs of mottling and mosaicism, which frequently disappear in the mature foliage. Ruiz et al. [35] conducted a study in which various germplasm samples of *Cucumis melo* were subjected to mechanical infections along with those of other species such as *C. anguria*, *C. ficifolius*, *C. myriocarpus*, and *C. metuliferus*. These strains were infected with European and Asian CGMMV strains. Resistance was gauged by observing symptom severity and analyzing viral accumulation by qRT-PCR. Results revealed that *C. anguria* and *C. ficifolius* remained symptom-free and did not accumulate CGGMV after inoculation. Conversely, *C. metuliferus* was markedly susceptible to isolates from both CGMMV strains. Although the virus was detected in *C. myriocarpus*, only the European isolate caused symptoms, whereas the Asian isolate did not. All 30 *C. melo* samples were vulnerable to CGMMV infection. Moreover, 16 melon samples displayed varying degrees of symptoms based on the isolate: milder symptoms with the European variant and more pronounced symptoms with the Asian strain at 14 and 21 d post-inoculation, respectively. Most germplasm samples exhibited significantly elevated viral concentrations, making them potential candidates for breeding initiatives [35].

Baudracco-Arnas and Pitrat, ref. [36] constructed a genetic map of the melon genome, spanning 1390 cM across 14 linkage groups. This comprehensive map was established using a diverse array of markers, including thirty-four Restriction Fragment Length Polymorphisms (RFLPs), sixty-four Randomly Amplified Polymorphic DNAs (RAPDs), one isozyme, four markers associated with disease resistance, and a single morphological marker. The data for this map were collected from 218 F_2_ plants, originating from a cross between ‘Védrantais’ and ‘Songwhan Charmi.’ Remarkably, among these markers, a gene named *nsv* was identified, conferring resistance to melon necrotic spot virus [36]. Subsequently, Morales et al. [37] mapped the melon necrotic spot virus gene nsv, identifying the flanking markers OPD08-0.80, CTA/ACG115, and CTA/ACG120 at 4.4 and 1.5 cM. From a population of PI 161375 (Pinyonet Piel de Sapo), two flanking markers for the nsv gene, X15L and M29, were discovered at 0.25 and 2.0 cM, respectively, providing resistance against melon necrotic spot virus [37].

Saez et al. [38] used RNA sequencing to investigate the genes associated with resistance to tomato leaf curl New Delhi virus (ToLCNDV) in *Cucumis melo*. By analyzing the transcriptomes of two genotypes, WM-7 (resistant) and Piñonet Piel de Sapo (PS) (susceptible), they identified regions associated with ToLCNDV-ES resistance. The primary resistance locus in WM-7 was mapped to Chr. 11, spanning positions 30,112,560, 30,737,924 bp, whereas secondary loci were found on Chr. 2 and 12. Within major Chr. 11, and three differentially expressed genes (DEGs) that potentially play key roles in the resistance mechanism were identified. These genes included *MELO3C022327.2*, a putative transmembrane protein; *MELO3C022337.2*, an auxin-responsive protein *SAUR36*; and *MELO3C022319.2*, the DNA primase large subunit. These results offer valuable insights into the molecular mechanisms underlying ToLCNDV resistance in melons [38]. Roman et al. [39] studied the molecular underpinnings of ToLCNDV resistance in *C. melo* subsp. *agrestis* group Momordica. Specifically, they focused on genotypes PI 414723 (Mom-PI414Ind) and PI 124112 (Mom-PI 124Ind). Through transcriptional analysis, they examined 10 candidate genes believed to be related to either proviral mechanisms, which facilitate viral infection, or antiviral mechanisms, which confer resistance. Notably, one proviral factor gene, *ARP4*, exhibited distinct differences in expression between the susceptible and resistant melon accessions, suggesting its potential role in modulating ToLCNDV resistance [39].

Using a high-resolution mapping approach, Amano et al. [40] identified a candidate gene for zucchini yellow mosaic virus (ZYMV) resistance in cucumbers, using MAS for the development of resistant cucumber cultivars [40]. The *zym* locus was mapped to Chr. Six from two closely linked SSR markers and identified a candidate gene at the *zym* A192-18 locus for ZYMV resistance in cucumbers. However, melon PI 414723 displayed resistance to ZYMV, which was attributed to the dominant allele at the *Zym* locus. This specific resistance trait hinders the spread of the virus within plants. Therefore, although typical mosaic symptoms are absent, plants may sometimes exhibit necrotic patches in response to viral inoculation. Earlier research identified the location of the *Zym* resistance gene in the second linkage group of the melon genetic map [41]. Adler-Berke et al. [41] identified the *Zym* locus responsible for ZYMV resistance in melon by cloning it from the CM-AG36 SSR marker located approximately 2 cM away. Using five mapping populations with the common resistant parent, PI 414723, they narrowed down the *Zym* locus to an interval of approximately 18 kb. They further constructed a BAC contig, isolating a BAC clone from the ZYMV-susceptible melon type MR-1 that encompasses the resistance gene. Another clone was extracted from the susceptible variety WMR 29. After sequencing and annotating both clones, they identified a specific 17.6 kb region that houses an NAC-like transcription factor and, depending on the variety, two or three R-gene analogs characterized by a CC-NBS-LRR structure. Subsequent mapping with SNP markers confirmed these findings. Despite identifying multiple polymorphic sites across different varieties, the team was unable to associate resistance with a single DNA polymorphism. However, their work has significantly advanced our understanding of the *Zym* locus, setting the stage for in-depth functional studies of this critical gene [41]. Plants expressing the full-length *CP-ZYMV* gene were extremely resistant; however, expression of the amino-truncated core region or antisense *CP-ZYMV* provided minimal protection [42]. This type of resistance depends on the action of ribozymes and small RNA molecules and their capacity to specifically cleave viral RNA. SSR markers have also been developed for the selection of ZYMV-resistant melon varieties [43].

#### 3.1.2. Fungal Pathogens

Fusarium wilt, which is induced by the fungal pathogen *Fusarium oxysporum*, is one of the most damaging soil-borne diseases affecting cucurbits. It was reviewed by Oumouloud et al. [44] that the disease is caused by *F. oxysporum* f. sp. *melonis* (Fom) in melons. Four physiological races, designated 0, 1, 2, and 1,2 were created from the Fom-1 and Fom-2 isolates, which are the two dominant resistance genes that regulate resistance to races 0 and 2, and 0 and 1, respectively. Melon lines with *Fom-1* and *Fom-2* resistance genes can develop infections upon exposure to Fom races 1, 2, and 3. The cloned *Fom-2* gene encodes a protein with leucine-rich repeats (LRRs) and nucleotide binding site (NBS) domains. Using composite interval mapping, nine QTLs for Fom races 1 and 2 were identified in five LGs. These QTLs may be used in combination with important resistance genes in other races to transfer resistance to high-yielding genotypes [44]. SSR markers, including *DM0096* and *CSWCTT02*, have been developed to effectively select *Fusarium* wilt-diseased plants [45]. Molecular markers, including AFLP, RAPD, and RFLP, linked to resistance to *Fusarium* race 2, have been developed for the selection of disease-resistant varieties [46].

Gummy stem blight (GSB) is a severe disease caused by the ascomycete fungus *Didymella bryoniae* (Auersw.) Rehm and its anamorph *Phoma cucurbitacearum* (Fr.) Sacc., which significantly reduces melon yield worldwide. Symptoms of GBS include the growth of necrotic lesions on leaves, water-soaked hypocotyls and leaves. These actions lead to the formation of stem cankers in the cortical tissue, resulting in a distinctive brown gummy exudate. Lesions continue to develop under susceptible interactions, eventually girdling the stem and causing wilting and plant death [47]. Currently, melon cultivars carry a single GSB resistance gene, which is insufficient to safeguard plants against a wide variety of *D. bryoniae* isolates. Genetic analysis revealed resistance to the Gsb pathogen as determined by independent genetic loci. Specifically, five dominant resistance loci (*Gsb-1*, *Gsb-2*, *Gsb-3*, *Gsb-4*, and *Gsb-6)* were identified across the PIs PI140471, PI157082, PI511890, PI482398, and PI420145. In contrast, a recessive resistance locus, *gsb-5*, has been identified in PI482399 [19,48]. Hassan et al. [49] evaluated 60 melon genotypes, including 16 lines and 44 cultivars, by using 20 SSR markers. They discovered that the SSR marker ‘C*MCT505*’, linked to the *Gsb1* gene on Chr.1, effectively distinguished between resistant and susceptible lines, suggesting that resistance is likely associated with the *Gsb1* gene. Analysis of the *Gsb1* gene sequences from both resistant and susceptible varieties revealed a 32 bp deletion in resistant lines and a 39 bp deletion in resistant cultivars, as compared to the susceptible ones [49]. Hu et al. [50] developed a genetic map using 12,932 markers spanning 1818 cM with an average marker distance of 0.17 cM. They determined the potential *Gsb* resistance genes located within a 108 kb section of pseudo-Chr. 4 [50]. Their genetic study of lines that were either resistant or susceptible to Gsb showed that the resistance was driven by a single dominant gene. Muskmelon Baipicui was hybridized with the donor parent 4598, which carried *Gsb-4* and *Gsb-6*. Field research has revealed that individuals with both *Gsb-4* and Gsb-6 exhibit increased GSB resistance and yield superior fruit quality [51]. Hassan et al. [52] identified a GSB resistance single recessive gene (*MELO3C022157*) encoding the nucleotide binding site–leucine-rich repeat (*TIR-NBS-LRR*) from the F_2_ resistant and susceptible lines developed from a cross between the ‘Cornell ZPPM 339’ (GSB-susceptible) and ‘PI482399’ (GSB-resistant) lines located on Chr. 9 in melon. Further cloning and sequencing of the TIR-NBS-LRR-type resistance gene exhibited a 2 bp deletion in the second exon and a 7 bp insertion in the fourth exon of the resistant line. In addition, two insertion/deletion (InDel) markers, *GSB9-kh-1* and *GSB9-kh-2*, were located in the first intron of *MELO3C022157* and were associated with GSB resistance. These markers were validated in the F_2_ population and inbred lines, indicating their potential for marker-assisted selection for breeding GSB-resistant melon varieties [52].

Powdery mildew (PM) is a deadly fungal disease that threatens the melon industry. The PM induced by *Podosphaera xanthii* (Castag.) U. Braun, N. Shish, and *Golovinomyces cichoracearum* (D.C.) Huleta is an established threat to melon production in all growing regions. Pathogenic infections impair leaf photosynthetic activity and reduce fruit yield and quality. However, chemical treatment can also be used to suppress this disease [53,54]. Resistant cultivars are preferred for financial and safety reasons. Cao et al. [14] employed bulked segregant analysis (BSA) and next-generation sequencing (NGS) to map powdery mildew resistance genes in an F_2_ segregating population resulting from a hybrid between a resistant (wm-6) and susceptible cultivar (12D-1) of melon. On Chr. 12, a new quantitative trait locus (QTL) named *qCmPMR-12*, showing resistance to PM, was identified ranging from 22.0 Mb to 22.9 Mb. RNA-Seq analysis revealed that the *MELO3C002434* gene, encoding an ankyrin repeat-containing protein, was the most probable gene linked to PM resistance. To date, four *P. xanthii* resistance genes and four PM resistance QTLs have been identified [14]. Wang et al. [10] identified a major QTL for PM resistance; the QTL is strongly associated with the SSR marker *CMBR120* [10]. Several types of PM resistance have emerged in the USA, using exotic accessions from India. Using whole-transcriptome sequencing, it was possible to detect circRNAs and ascertain their molecular roles in the leaves of PM-resistant (M1) and -susceptible (B29) melons. Over 50% of the 303 circRNAs discovered originated from exonic regions. After PM infection in B29 and M1 cells, 17 and 23 circRNAs showed significantly different expression levels, respectively. Based on the functional annotation of circRNAs that regulate parental genes, melon circRNAs may participate in responses to biotic stimuli, oxidation–reduction, metabolic activities, and regulation of gene expression [55]. Competitive allele-specific PCR (KASP) technology has been introduced to confer PM resistance to melons [13,14]. Zhang et al. [56] analyzed the melon genome and identified 14 MLO gene family members across eight chromosomes. They categorized these genes into five groups based on their phylogenetic relationships and found that genes within the same group shared similar intron and exon patterns. Furthermore, they cloned the *CmMLO* gene from four melon varieties and examined its post-infection expression. They focused on the *CmMLO5* gene (*MELO3C012438*), which encodes (MLO-like protein) in the susceptible Topmark melon variety, noting its detrimental effect on resistance to PM. Key observations were made at 24 and 72 h after PM infection. Mutational analysis revealed a single mutation at 572 bp, leading to the loss of protein function and, consequently, increased resistance to PM [56].

Fusarium wilt, gummy stem blight, and powdery mildew are fungal diseases that pose severe threats to melon crops. Recent studies have enhanced our understanding of the genetic underpinnings of these diseases, with various genes and QTLs identified as governing resistance. Advanced genetic mapping, next-generation sequencing, and innovative techniques have propelled the development of resistant melon varieties. These breakthroughs offer promising pathways to bolster melon crop resilience, ensure better yields, and minimize the need for chemical interventions.

#### 3.1.3. Bacterial Pathogens

Bacterial fruit blotch (BFB) is a disease caused by the aerobic, Gram-negative, rod-shaped, seed-borne bacterium *Acidovorax citrulli* (formerly *A. avenae* subsp. *citrulli*), which causes considerable financial losses to melon crop growers worldwide. Although this bacterium reduces the amount of marketable melons produced and affects the farming economy, the genetic loci that confer disease resistance and the mode of inheritance have remained unknown until recently. Host resistance is the most economical and eco-friendly method for managing BFB [57].

Islam et al. [58] investigated the genetics of BFB resistance in melons. Through research on a sample of 491 F_2_ individuals, they deduced that BFB resistance is controlled by a single dominant gene. This was inferred from the observed inheritance pattern, which showed a 3:1 segregation ratio between resistant and susceptible phenotypes. The population for this study was derived from a cross between a BFB-resistant parent (PI 353814) and a BFB-susceptible parent (PI 614596). Of the 57 observed potential disease resistance genes spread across the melon genome, only the *MELO3C022157* gene, which encodes the *TIR-NBS-LRR domain*, exhibited a notable difference between the resistant and susceptible parents. To investigate this, cloning, sequencing, and quantitative RT-PCR analysis of the polymorphic *MELO3C022157* found on Chr. 9 were performed. Detailed examination revealed several insertion/deletion mutations (InDels) and SNPs. Notably, an A2035T SNP in the second exon of the gene resulted in the loss of the LRR domain in the susceptible variant, leading to a truncated protein. This pivotal identification underscores the significance of this specific mutation in influencing BFB resistance in melons [58]. Islam et al. [58] developed the PCR-based co-dominant insertion-deletion (InDel) marker MB157-2 to distinguish between resistant and susceptible accessions and demonstrated 98.17% and 100% detection accuracies, respectively. Marker development relied on a significant (504 bp) insertion found in the first intron of the susceptible accession. This marker effectively differentiated between resistant and susceptible accessions within a cohort of 491 F_2_ individuals and 22 landrace/inbred accessions. The development of this novel PCR-based co-dominant InDel marker has useful ramifications for marker-assisted breeding to increase BFB resistance in melons [58]. In a subsequent study by Islam et al. [59], they studied the genetic mechanisms underlying bacterial fruit blotch (BFB) resistance in melon. This study involved the thorough characterization of the complete set of 70 R-genes (resistance genes) within the melon genome. This comprehensive analysis explored aspects such as gene structures, specific locations on chromosomes, the organizational patterns of their domains, the distribution of motifs, and synteny relationships. During their investigation, the team discovered several domains in these genes that are typically associated with disease resistance, including NBS, TIR, LRR, CC, RLK, and DUF. Based on the unique domains encoded by these proteins, R-genes were subsequently categorized. A pivotal part of their study was the gene expression analysis across different melon accessions. This exploration revealed that six R-genes (*MELO3C023441*, *MELO3C016529*, *MELO3C022157*, *MELO3C022146*, *MELO3C025518*, and *MELO3C004303*) exhibited distinct expression patterns. More specifically, these genes demonstrated increased expression in BFB-resistant melon variants compared to their susceptible counterparts. The significance of these findings cannot be overstated because these specific genes, once functionally validated, are prime candidates for targeted breeding programs and biotechnological interventions. Leveraging these genes has the potential to improve BFB resistance in melon cultivars, paving the way for more resilient crops in the future [59]

The groundbreaking research on bacterial fruit blotch (BFB) resistance in melons revealed the genetic foundations of the disease. The identification of key mutations influencing bacterial fruit blotch (BFB) resistance and the establishment of reliable PCR-based markers represent invaluable tools for future breeding programs. Furthermore, the comprehensive analysis of resistance (R)-genes in the melon genome, along with the identification of six distinct genes showing heightened expression in BFB-resistant melons, provides insights into potential pathways to fortify melon crops against this detrimental disease. This study sets the foundation for the development of more resilient melon cultivars, ensuring improved yields and reduced losses to farmers worldwide.

#### 3.1.4. Nematodes

Root-knot nematodes (RKNs) are a significant constraint on melon production in the southern United States and other semi-tropical and tropical regions worldwide. *Meloidogyne incognita*, *M. javanica*, and *M. arenaria* are the most common RKN species in the southeastern United States, and are responsible for causing root gal disease in melons. Many crop species, including melons and cucumbers, are susceptible to severe yield loss caused by *Meloidogyne incognita*. Interspecific hybridization between the wild cucumber species *Cucumis hystrix* Chakr. (2n = 24, HH), and cucumbers produced the introgression line IL10-1, which exhibited resistance against roots, but not nematodes. Based on low-coverage sequences of the F_2:6_ recombinant inbred lines (RILs) generated by crossing the inbred line IL10-1 and cultivar ‘Beijingjietou’ CC3, an ultra-high-density genetic linkage bin-map made up of high-quality SNPs was created and three QTLs *qRKN1-1*, *qRKN5-1*, and *qRKN5-2* were identified. The possible RKN resistance-related genes were proposed to be four genes with non-synonymous SNPs present on Chr. 5 [60].

RKNs pose a significant threat to melon production. Recent studies have leveraged an interspecific hybridization approach, yielding the introgression line IL10-1, which has shown nematode resistance. With advanced genetic mapping using SNPs, three QTLs associated with RKN resistance were identified, highlighting the potential of genomic interventions to enhance crop resilience to RKNs in melons.

#### 3.1.5. Insects and Herbivores

*Aphis gossypii,* or the melon aphid, is a major pest for melon crops, causing direct damage by feeding on sap and spreading plant viruses. To address this, researchers have developed melon varieties with resistance to these aphids using traditional methods, marker-assisted selection, and genetic engineering, ensuring better protection against *Aphis gossypii*. *Aphis gossypii* resistance in melon is due to the single dominant virus aphid transmission (Vat) gene, belonging to the nucleotide binding site–leucine-rich repeat family, which triggers significant transcriptional changes during aphid infestation [61].

Sattar et al. [61] explored the response of aphid herbivory in resistant and susceptible melon interactions using MicroRNAs (miRNAs). They constructed small RNA (smRNA) libraries from bulked leaf tissues of a Vat+ melon line during early and late aphid infestations. Sequence analysis showed that the expression profiles of both conserved and newly identified miRNAs changed at different stages of aphid herbivory, confirmed by quantitative PCR in both resistant Vat+ and susceptible Vat– interactions. Most conserved miRNA families were differentially regulated during early aphid infestation stages in both interaction types. Additionally, 18 cucurbit-specific miRNAs were expressed during various stages of aphid herbivory. This study provides insights into miRNA-dependent gene regulation in Vat-mediated melon resistance [61].

### 3.2. Abiotic Stress

Melon, a widely cultivated crop, is vulnerable to several abiotic stressors such as cold/chilling, drought, salinity, and temperature fluctuations. These pressures have a negative influence on melon development, yield, and fruit quality. Scientists are currently employing new breeding techniques to address these issues. Molecular markers have emerged as effective tools for this endeavor, allowing precise and effective breeding for abiotic stress resistance. The integration of molecular markers into breeding programs has the potential to accelerate the production of melon cultivars with an increased tolerance to harsh environmental conditions. This study aimed to provide insights into the most recent advances in marker-assisted breeding specifically tailored to mitigate the effects of abiotic stress in melons. Our ultimate goal was to pave the way for sustainable melon cultivation in the context of climate change.

#### 3.2.1. Cold/Chilling

Low temperatures are a significant abiotic stress that adversely affects the morphological growth and fruit development of melons. Lipoxygenases (LOXs) play crucial roles in abiotic stress responses, such as injury, UV rays, extremely high or low temperatures, oxidative stress, and drought, and understanding the activity of the LOX gene promoter may help to better understand the mechanisms underlying resistance. Hou et al. [62] analyzed an extended melon core sample of 212 distinct accessions using 272 SSRs and 27 CAPSs for freezing tolerance at the seedling stage by association mapping. Two primary groups of *C. melo*—*melo* and *agrestis*—were identified via genetic diversity analysis of the entire accession panel, among which *agrestis* was the most tolerant to chilling stress. The two subspecies and the entire panel were subjected to genome-wide association analysis (GWAS), which identified 51 loci involved in 74 marker–trait associations. A total of 35 of these associations were identified in the entire session panel: 21 in *melo* and 18 in *agrestis*. *CMCT505_Chr.1* was repeatedly found across numerous populations, with significant phenotypic contributions, and may be a crucial locus regulating *C. melo’s* chilling tolerance [62].

The *CmLOX08* promoter contains a large number of cis-regulatory components linked to signaling molecules and abiotic stress. In melon, the *CmLOX08* promoter (2054 bp) was cloned using the *E. coli* strain DH5α [63]. By analyzing different sections of this promoter, distinct regions regulating its activity in response to various signals and stresses were identified. Different segments were found to be central to the temperature responses, with specific regions amplifying or inhibiting promoter expression. Intriguingly, while the segments from −2054 to −1284 bp negatively influenced promoter activity under heat stress, the region spanning from −284 to −1 bp emerged as a core sequence that actively responded to heat. In terms of cold stress, the main responsive segment was identified as −1047 to −1 bp, while the region from −2054 to −1047 bp acted to suppress the promoter’s expression under cold conditions. The authors noted that after a 6 h exposure to cold stress, low temperature dramatically elevated the expression of *CmLOX08*. These results provide insights into the intricate regulation of the *CmLOX08* promoter under temperature stress [63]. Zhang et al. [64] performed research on Gold Queen Hami melon (susceptible to cold damage) to explore gene expression during cold exposure. The two cold treatment durations (12 and 24 days) were compared to the controls, which showed thousands of differentially expressed genes (DEGs) due to cold stress. Key cold-responsive genes emerged from pathways involving starch and sucrose metabolism and glycolysis. They also identified significant transcription factors and verified these findings using qRT-PCR. To confirm the RNA-seq findings, 12 genes were randomly selected for qRT-PCR. The qRT-PCR results aligned well with the RNA-Seq data, verifying the accuracy of the transcriptome analysis. These insights pave the way for enhancing cold resistance in Hami melons [64].

Song et al. [65] identified 39 glutathione S-transferase (GST) genes, which are essential for combatting oxidative stress, and grouped them into seven subfamilies. They observed that the cold-tolerant variety, Jia Shi-310, displayed enhanced GST activity and increased expression of 28 cold-stress genes. In particular, *CmGSTs* from the Tau, Phi, and DHAR categories were vital in the cold response, marking them as potential candidates for in-depth research. Their study broadens the knowledge of the role that GSTs play in Hami melon stress adaptation and how *CmGST* genes contribute to Hami melon cold tolerance mechanisms [65]. Later, Li et al. [66] observed a balance between seedling cold tolerance and fruit quality in melons and examined primary metabolites in eight melon lines with varying cold tolerances. Cold-resistant melons generally had lower metabolite concentrations than cold-sensitive melons. The most significant difference was found between the cold-resistant H581 and moderately cold-resistant HH09 lines. Using these data, five pivotal genes responsible for balancing cold tolerance and fruit quality were identified. One specific gene, *CmEAF7*, was highlighted for its potential to regulate various growth processes and improve both cold tolerance and fruit quality, and offers a promising direction for breeding improved melon varieties [66].

#### 3.2.2. Drought

Plants express many proteins under drought stress that are not explicitly connected to drought but are instead induced as a result of cellular damage. These are induced by genes associated with heat shock proteins (HSPs), proteinase inhibitors, thiol proteases, and osmotin. Ansari et al. [67] studied the responses of two muskmelon genotypes (drought-tolerant SC-15 and drought-susceptible EC-564755) to progressive water stress. The authors observed a higher activity of stress enzymes and higher expression of *DREBs, RD22*, and *dehydrin* genes in the drought-tolerant SC-15 genotype [67]. Subsequently, Ansari et al. [29] performed a study using the drought-resistant muskmelon genotype SC-15 and found that drought conditions led to decreased photosynthesis, stomatal conductance, and transpiration rates, along with reduced chlorophyll and carotenoid content. However, the plant responded by increasing its defense mechanisms, as evidenced by an increase in antioxidant enzyme activities and changes in protein abundance related to various cellular processes. Specifically, 38 proteins increased in abundance, whereas 10 decreased. The identified proteins belonged to various functional groups including protein synthesis, nucleotide biosynthesis, photosynthesis, metabolism, stress response, transcriptional regulation, and DNA binding. A drought-induced MADS-box transcription factor was also observed, highlighting the efforts of plants to enhance defense and reduce destructive processes under drought stress [29]. Xing et al. [68] observed increased expression of the *CmLOX10* gene in oriental melon leaves under drought stress. Furthermore, it has been reported that reducing *CmLOX10* made melons more drought-sensitive, whereas increasing it in *Arabidopsis* plants enhanced their drought resistance through genetic modifications. This change was linked to the jasmonic acid (JA) pathway; higher *CmLOX10* levels led to increased JA, promoting drought tolerance and causing stomatal closure. Additionally, JA influenced *CmLOX10* through a feedback loop involving *CmMYC2*, which directly interacted with the *CmLOX10* promoter. *CmLOX10* plays a key role in the drought response of melons, primarily through JA-mediated mechanisms involving stomatal control and feedback with *CmMYC2* [68].

Briefly, under drought stress, muskmelon plants activate a range of protective mechanisms, including the expression of proteins not directly related to drought. These key findings highlight the heightened stress responses in the drought-tolerant SC-15 genotype and the pivotal role of the *CmLOX10* gene in drought resistance. The interaction of this gene with the jasmonic acid pathway, particularly through feedback mechanisms involving the *CmMYC2* gene, underscores the complex interplay of mechanisms that enhance plant drought resilience.

#### 3.2.3. Salt

Melons are negatively affected by salt stress, which can lead to accumulation of sodium ions, disruption of cellular balance, and dehydration. Stress can hinder seed germination, growth, and fruit development. In addition, salt exposure can elevate oxidative stress in melons, potentially damaging cellular and metabolic functions. In addition, salt stress triggers molecular responses that alters gene expression patterns, leading to the upregulation of salt-responsive genes in melons. These genes encode proteins involved in sodium ion transportation, osmotic balance, and detoxification pathways. Moreover, salt exposure can initiate signaling pathways that enhance the production of antioxidants to counteract the oxidative stress induced by excess salts. At the molecular level, this stress can influence the functionality of key enzymes and DNA methylation patterns and impact various metabolic pathways critical for melon growth and fruit development.

To explore how muskmelons deal with salt stress, researchers have created a cDNA library from the salt-tolerant variety Bingxuecui. The authors identified 312 quality expressed sequence tags (ESTs) in 339 clones. The analysis showed that many of these ESTs resembled genes known for stress responses in plants. These genes have various functions, including metabolism, energy, and defense, indicating a multifaceted response to salt stress in melons. Real-time PCR tests of 27 ESTs confirmed their activation under salt stress, suggesting the need for in-depth studies to understand the broader implications of these genes under high-salinity conditions [69]. NAC proteins in plants are crucial for development and stress responses. In melon, 82 NAC genes were identified. By analyzing melon data under salt stress and comparing them with known *Arabidopsis* NAC genes, a specific group of melon NAC genes was found to be vital for the salt stress response. Eleven out of twelve of these genes were triggered by salt, with one exception. *CmNAC14* expression increased after 12 h of salt exposure. When introduced into *Arabidopsis,* this gene makes the plants more salt sensitive, as evidenced by altered physiological metrics and decreased stress response gene activity. This study sheds light on how melon NAC genes influence salt stress responses [70]. According to Shah et al. [71], the multidrug and toxic compound extrusion (MATE) protein family plays an important role in plant defense by exporting toxins and performing other functions. In an extensive study on four Cucurbitaceae species, 174 MATE genes were identified. These genes were classified into seven subgroups and were mostly found in the plasma membrane. Transposed duplication events influence the gene family expansion in *Cucumis melo*, implying that different gene architectures indicate different activities. These genes are activated by several transactivating factors under stressful conditions. The chromosomal location, molecular traits, and relationships with other species have been studied, revealing a high number of orthologous genes. MiRNAs have been found to regulate *MATE* genes, and certain genes exhibit different expression levels under salt stress. Furthermore, a specific *MATE* gene from *Cucumis melo* showed that its protein localized to the plasma membrane in *Arabidopsis thaliana*. The present study sheds light on the *MATE* gene family’s role in Cucurbitaceae, particularly under salt stress conditions [71]. Chevilly et al. [72] performed a study of two different melon varieties comparing sensitive and tolerant cultivars and reported that varieties or cultivars with increased histidine content and/or the ability to accumulate sodium Na+ (likely in the vacuoles) may display improved tolerance to salt stress [72].

#### 3.2.4. Toxic Compounds

The family of proteins known as multidrug and toxic compound extrusion (MATE) proteins mediates detoxification in plants. However, data on similar MATE proteins in melons were unavailable until recently. Wang et al. [73] identified 39 *CmMATE* proteins by genome-wide characterization of the MATE family in the melon genome to generate abiotic stress-resistant melon variants [73]. Recently, Branham et al. [23] developed KASP markers to assess the tolerance of melons to sulfur phytotoxicity. Elemental sulfur, an effective fungicide, can be toxic to many melon varieties, causing leaf damage and even plant death. To understand this, a GBS-based genetic map of RIL was constructed to study sulfur tolerance in melons. Three key regions (QTLs) linked to sulfur tolerance were identified on Chr. 1. One major QTL region, *qSulf-1*, was further narrowed down, and a gene *MELO3C024245* encoding sorting nexin-13 was identified, which might be involved in sulfur tolerance in melon. Subsequently, it was validated using KASP markers that were verified in various melon cultivars. This study provides a tool for breeding sulfur-tolerant melon varieties [23].

Zhang et al. [64] examined the MYB transcription factors in melons, which are vital for stress responses, using genome-wide analysis. In total, 178 MYBs were identified, of which 81 belonged to the 1R-MYB subfamily and 95 belonged to the R2R3-MYB subfamily. These proteins are mostly located in the nucleus and are distributed across all melon chromosomes. Their evolution appears to be driven by tandem and fragment duplications, which are especially prominent in Chr. 4. MYBs contain elements that respond to stress, light, and other hormones. Under autotoxic conditions, 60 MYBs showed varied expression, whereas 6 reacted to saline–alkali stress. Notably, *CmMYB2R40* and *CmMYB2R54* respond to both autotoxicity and saline–alkali stress, potentially influencing osmotic regulation and other processes, offering insights into MYB’s role in melon stress resistance [64].

Advancements in genomics have significantly expanded our understanding of the mechanisms underlying melon resistance to abiotic stressors. The identification of MATE proteins in melons has opened pathways for the development of variants that are resistant to external stress. Research on sulfur toxicity has led to the development of markers for breeding sulfur-tolerant melon varieties. Moreover, the discovery and classification of *MYB* transcription factors further emphasize the complexity of the stress-response mechanisms of melons. Taken together, these findings offer a promising direction for improving melon resilience under adverse environmental conditions. Detailed information on the candidate genes for biotic and abiotic stresses is presented in Table 2.

## 4. Molecular Markers for Marker-Assisted Backcrossing (MABC) and Cultivar Identification

Molecular markers play a crucial role in modern plant breeding and genetics and offer valuable insights into genome intricacies. Marker-assisted backcrossing (MABC) is a breeding technique that uses molecular markers to facilitate the selection of desirable traits from a donor parent by incorporating them into an elite or recurrent parent through backcrossing. This approach ensures the rapid integration of favorable traits while preserving the genetic integrity of elite cultivars. Molecular markers have become indispensable tools in contemporary plant breeding and genetics, particularly for MABC and cultivar identification. MABC utilizes specific markers associated with the desired traits to enhance traditional backcrossing by improving its accuracy, speed, and efficiency. The markers commonly used in this process include simple sequence repeats (SSRs), single nucleotide polymorphisms (SNPs), and random amplified polymorphic DNA (RAPD). For cultivar identification, a process crucial for ensuring the authenticity and purity of varieties, markers such as SSRs, SNPs, intersimple sequence repeats (ISSRs), and amplified fragment length polymorphisms (AFLPs) are preferred. These markers not only guarantee cultivar authenticity but also safeguard breeders’ rights and facilitate effective germplasm management. As advancements in technology continue, the role of molecular markers in crop improvement and management is poised to expand into promising, more efficient, and innovative approaches to plant genetics.

Modern molecular breeding techniques provide opportunities for targeted and efficient enhancement of melon cultivars. For instance, Sousaraei et al. [24] employed marker-assisted backcrossing to introduce the Fusarium wilt resistance gene Fom-2 from a resistant genotype (Isabelle) into two susceptible cultivars (Garmak and Tile-Torogh). In their study, Sousaraei et al. [24] isolated, cloned, and sequenced the 1274 bp of the 3′ end of the leucine-rich repeat (LRR) domain of Fom-2 from both resistant and susceptible genotypes. Notably, they observed 28 nucleotide substitutions between the resistant and susceptible genotypes within that region. This research highlights the potential of molecular breeding techniques to transfer resistance traits efficiently, providing a promising avenue for the improvement of melon cultivars [24]. Specific primer pairs, Fom2-R409 and Fom2-S253, were designed based on allele substitutions to amplify genes associated with resistance and susceptibility, respectively. Resistant plants in the BC_1_F_1_ and BC_2_F_1_ generations were initially identified through artificial pathogen inoculation and their resistance was later validated by genotyping using functional markers. The selection of resistant plants was also based on the phenotypic characteristics in each generation, which led to a high degree of similarity between the backcross 3 (BC_3_) generation and recurrent parents. This study demonstrated that the newly developed markers exhibited higher accuracy and efficiency than the inoculation trials, establishing them as dependable tools for screening Fusarium wilt-resistant genotypes in melons. Olalekan et al. [76] employed marker-assisted backcrossing (MABC) to create a Fusarium wilt-resistant watermelon variety by integrating wilt resistance genes from the resistant inbred line CS-19 into the genome of the high-yielding, wilt-susceptible inbred line BL-14. Furthermore, they tested 380 SSR markers to identify polymorphisms among the parents, with 78 markers showing variations. Backcross analysis revealed a recovery range of 74.7% to 94.4% in the BC_1_F_1_ generation and 86.6% to 96.8% in the BC_2_F_1_ generation [76].

## 5. Limitations and Challenges

Molecular markers have brought about significant advancements in plant breeding, offering enhanced efficiency in selecting desired traits. In the context of marker-assisted breeding (MAB) for biotic and abiotic stress resistance in melons, distinct limitations and challenges must be considered. The melon genome is intricate and often presents challenges due to its polyploid nature, which can confound the identification and mapping of markers associated with specific traits. Moreover, the vital step in the phenotypic verification of a trait, even after its marker has been recognized, requires time and controlled environmental conditions, making it a laborious endeavor. Some stress resistance genes in melons may not have well-mapped or reliable markers, which poses further challenges.

Complexities also arise because many of these resistances are controlled by multiple genes whose interactions or epistasis can impact the manifestation of the trait. Environment plays a pivotal role in trait expression, and its variability can lead to inconsistencies in the association of markers in diverse settings. However, the financial and logistical barriers should not be overlooked. Despite becoming more affordable over the years, genotyping remains expensive for expansive breeding programs. In addition, cutting-edge molecular techniques may be out of the reach of breeders in areas with limited resources. A phenomenon called linkage drag can also pose challenges, in which undesired traits linked to beneficial ones are inadvertently selected.

From a technical standpoint, the specificity, sensitivity, and overall capacity of molecular techniques are not always consistent, leading to erroneous selection due to false positives or negatives. Owing to the burgeoning amount of data in MAB, there is a pressing need for sophisticated database systems and bioinformatic tools for data management and interpretation. Challenges are not limited to the plant genome; evolving populations of pathogens mean that a resistance gene effective today might be rendered ineffective if the pathogen mutates. There is prevalent confusion between MAB and genetic modifications. Although MAB is an integral aspect of traditional breeding, it might be prone to misconceptions and encounter regulatory and acceptance issues. In the broader context, to realize the complete potential of molecular markers in melon breeding, a collaborative approach involving molecular biologists, breeders, and other stakeholders is crucial.

## 6. Conclusions and Future Perspectives

In recent years, the integration of molecular markers into melon breeding practices has sparked a revolution in crop improvement strategies. These markers empower breeders to precisely identify and incorporate genes responsible for resistance to both biotic and abiotic stresses, thereby enhancing the resilience and productivity of melon crops. This advanced approach, known as marker-assisted selection (MAS), facilitates systematic improvements in breeding cycles, reduces dependence on conventional phenotypic evaluations, and ensures precise genetic modifications. As the landscape of genomics and bioinformatics continues to evolve, the future of MAS in melons is poised for significant expansion. There is a critical need to identify novel markers linked to pivotal stress resistance traits. Moreover, the convergence of MAS with innovative technologies, such as the CRISPR-Cas system, holds promise for refined genomic editing without overhauling the plant’s genetic structure. Given the unpredictable nature of global climate patterns, melons will inevitably encounter evolving stress challenges, necessitating continuous adaptation and innovation of MAS techniques. The advent of high-throughput sequencing and phenotyping platforms is anticipated to deepen our genetic insights and enhance the accuracy of phenotypic evaluations. As data analytics become more sophisticated, breeders have the potential to leverage artificial intelligence to discern intricate genetic patterns, predict trait interplay, and devise optimal breeding paths. This advanced approach is expected to be enriched by collaborative efforts involving researchers, farmers, industry stakeholders, and policymakers, fostering a shared knowledge environment and facilitating global germplasm exchange. However, as we navigate this promising trajectory, it is crucial to remain cognizant of bioethical considerations, adhere to regulatory guidelines, and be attuned to public perceptions surrounding these cutting-edge breeding techniques. A balanced, technology-driven, and informed approach to the future of MAS in melons holds promise for significantly impacting global agricultural sustainability and food security.

## Figures and Tables

**Table 1 ijms-25-06307-t001:** Quantitative trait loci (QTLs) linked with biotic and abiotic stress in different melon cultivars.

S. No	Traits	QTL Name	Parents	Generation ^a^	Linkage Group	Markers ^b^	Reference
I	Biotic Stress						
1.	Powdery mildew	*qPM2*	TARI-08874xBai-li-gua	F_2_	II	CSJCT358, CMBR120	[10]
		*Pm-R*	TGR-1551 x Bola de Ora	F_2_	V	PM3-CAPS	[11]
		*BPm12.1*	MR-1xTop Mark	F_2_	XII	(CAPS) BSA12-LI3ECORI, BSA12-LI4HINFI	[12]
		*qPx1-5*	PI 124111x Ananas Yokne’am (AY)	RIL	V	(KASP) pm1-5_25329892, pm1-5_25461503 pm1-5_25625375	[13]
		*qPx1-12*			XII	(KASP) pm1-12_22848920 pm1-12_22904659	
		*CmPMR-12*	Wm-6x 12D-1	F_2_	XII	(KASP) KA002213 and KA002215	[14]
2.	Cucumber mosaic virus (CMV)	*CmVPS41*	Védrantais x PI 161375	RILs	NA	NA	[15]
		*A major QTL*	Piel de Sapo x PI 161375	NILs	XII	CMN61_44, CMN21_55	[16]
3.	*Fusarim* wilt	*qFom-1.2-9*	Védrantais x Isabelle.	RIL	IX	NA	[17]
		*qFom-1.2-11*			XI	NA	
		*qFom1-2*	MR1x Ananas Yok’neum	RIL	II	S2_15048162, S2_15110453	[18]
		*qFom1-7*			VII	S7_13949405, S7_13949474	
		*qFom1-11*			IX	S11_6921979	
		*qFom1-12*			XII	S12_18312135, S12_18322152, S12_18322266, S12_18380444	
4.	Gummy stem blight	*Gsb-1*	Cornell ZPPM 339’ x PI482399	F_2_	I	NA	[19]
		*Gsb-2*			II	NA	
		*Gsb-3*			III	NA	
		*Gsb-4*			IV	NA	
		*gsb-5*			V	NA	
		*gsb1.1*	JSD-3 x S717	F_2:3_	I	NA	[20]
		*gsb2.1*			II	NA	
		*gsb3.1*			III	NA	
		*gsb5.1*			V	NA	
		*gsb6.1*			VI	NA	
5.	Tomato leaf curl New Delhi virus (ToLCNDV)	*ToLCNDVVT30_2*	WM-7x Piel de Sapo	F_2_ and BC_1_	II	CMPSNP658	[21]
		*ToLCNDVSy15_11*			XI	CMPSNP475	
		*ToLCNDVSy30_11*			XI	CMPSNP475	
		*ToLCNDVVT30_11*			XI	CMPSNP475	
		ToLCNDVSy15_12			XII	AI_35-A08	
		*ToLCNDVSy30_12*			XII	AI_35-A08	
		*ToLCNDVVT30_12*			XII	AI_35-A08	
6.	Alternaria leaf blight (ALB)	*qALB-10*	MR-1x Ananas	RIL	X	S10_10324787	[22]
		*qALB-12*			XII	S12_22731131	
II	Abiotic stress						
1.	Sulfur tolerance	*qSulf-1*	MR-1x Ananas Yok’neum	RILs	I	Sulf1_33860724, Sulf1-33791317, Sulf1-33804906, Sulf1-33835488, Sulf1-33851209	[23]
		*qSulf-12*			XII	NA	
		*qSulf-8*			VIII	NA	

^a^ recombinant inbred line (RIL), near-isogenic line (NIL), and backcross (BC). ^b^ cleaved amplified polymorphic sequences (CAPSs), competitive allele-specific PCR (KASP), and not available (NA).

**Table 2 ijms-25-06307-t002:** Candidate genes linked with biotic and abiotic stress in different melon cultivars.

Traits	Locus Name	Gene ID	Gene Function	Chr.	Marker Name (Type ^a^)	Primer Sequence (5′–3′)	Ref.
**Bacterial fruit** **blotch**	NA	*MELO3C022157*	TIR-NBS-LRR domain	9	*MB157* (InDel)	F: ATGGAAGCAATTGAGGAATC R: TACAATGACCTAGTACTCCC	[58]
	NA	*MELO3C023441*	Receptor-kinase, putative	1	*M3441*	F: GGGAAAGAGTTAAATGCGAC R: CCATCTAGACCTTGGTTTCC	[59]
		*MELO3C016529*	TMV resistance protein N	6	*M6529*	F: CCGTGTGGCGGTCGGCGGTG R: ACAATGCCGCCACCGTCTTC	
		*MELO3C022157*	TMV resistance protein N-like isoform X1	9	*M2157*	F: GGAATCCATGGACGACGGAA R: TCCCTCCACCGATGAACCTG	
		*MELO3C022146*	TMV resistance protein N-like	9	*M2146*	F: GTACGGATGAACAAAAGCAT R: TCCATTGTTGAACCTCCTCC	
		*MELO3C025518*	Disease resistance protein RGA2-like	9	*M5518*	F: GGCACACGGTTTTCTTCAAC R: TTCTTCTATTCTTCTGGTCC	
		*MELO3C004303*	TMV resistance protein N-like	5	*M4303*	F: GGTTGGTGGACGTGATTGGT R: ACTTTCCTTAAAAGCATGCC	
**Zucchini** **Yellow Mosaic Virus**	*Zym*	*MELO3C015342*	Myb/SANT-like DNA-binding domain protein	2	NA	NA	[41]
		*MELO3C015353*	Disease resistance protein	2			
**Tomato leaf curl New Delhi virus (ToLCNDV)**	NA	*MELO3C022327*	Putative transmembrane protein	11	NA	F: CTTTCATCATGGTGTTCTCCGC R: AGAACAATCCTACCGTCGTTCC	[38]
		*MELO3C022337*	Auxin-responsive protein SAUR36			NA	
		*MELO3C022319*	DNA primase large subunit			F: AGTTGCTCGGTTGATTGGTC R: CGTCAGACTTAGGGCCTTTG	
**Powdery** **mildew**	NA	*MELO3C002434*	ankyrin repeats containing protein	12	NA	NA	[14]
	*O*	*CmMLO5* *MELO3C012438*	MLO-like protein	10	NA	F: ATGGCTGAATGTGGAACAGAGCA R: TCATTTGGCAAATGAGAAGTCCGA	[56]
	NA	*CmPMRl* *MELO3C002441*	Ankyrin repeat family protein	12	*PM12R-5* (CAPS)	F:GCCAACTAAGAGAATGTTCA R:AATGCTGGAGATGCTGTC	[74]
					*PM12R-6* (CAPS)	F:TATACCTCATCATCTCACTCC R:TGGTCGGTGTTGATACTAC	
		*CmPMrs* *MELO3C012438*	MLO-like protein	10	*PM10m-4* (CAPS)	F:TGGTTGAGAGCTTCATAGAT R:CCAAATGATTAGGTGTAGATGG	
					*PM10m-5* (CAPS)	F:CAATCCTGGCATACATTATCC R:CACTGTCACTATGGCTCAC	
**Cucumber mosaic virus (CMV)**	*cmv1*	*CmVPS41*	Vacuolar protein sorting 41	NA	NA	NA	[33]
**Gummy stem** **blight**	*Gsb5.1*	*MELO3C022157*	Nucleotide binding site–leucine-rich repeat	9	*Kh-GSB9-1* (Indel)	F: GTTAGGAAACAACAGACCTCCA R: CAGAACGCACAAAACTCAAAGGAC	[52]
					*Kh-GSB9-2* (Indel)	F: CCTAATAGTCCTTTGAGTTTTGTGCG R: GGTGTGCTTGGATTGGCTTTCT	
	*Gsb*	*MELO3C012987*	Encoding protein similar to the uncharacterized Avr9/Cf-9 rapidly elicited (ACRE) protein 146	4	NA	F: AGCAAGCTCATGGTTGATCTC F: GAGAAGGCGAGACCAGAGAC	[50]
	*Gsb-7(t)*	*MELO3C010403*	Putative receptor-like protein kinase	7	NA	NA	[75]
**Chilling/Cold tolerance**	*CMCT505*	*MELO3C013397*, *MELO3C013398*, *MELO3C013399*, *MELO3C013400*, *MELO3C013401*, *MELO3C013402*, *MELO3C013403*	SAUR-like auxin-responsive family protein	1	NA	NA	[62]
		*MELO3C013404*	Cellulose synthase catalytic subunit				
		*MELO3C013405*	Methionine-tRNA ligase				
		*MELO3C013406*	tRNA-dihydrouridine(47) synthase [NAD(P)(+)]				
		*MELO3C013407*	Putative transcription factor				
		*MELO3C013408*	Putative transcription factor PosF21				
		*MELO3C013409*	GAGA-binding transcriptional activator				
		*MELO3C013410*, *MELO3C013411*	NADP-dependent D-sorbitol-6-phosphate dehydrogenase				
	NA	*CmEAF7* *MELO3C012147.2*	Subunit of nucleosome acetyltransferase of H4 complex (NuA4)	10	NA	F: TCTGAGCTCTCTAGAATGGAAA R: TTTGGCGTCTTCCATAGACTCT	[66]
**Drought** **tolerance**	NA	*CmLOX10*	NA	5	NA	F: TGACAGGACAAGGAGTTC R: CGGTATTGGCAAGAATGTTA	[68]
**Salt and Drought** **tolerance**	NA	*CmLOX08* *MELO3C011885*	Lipoxygenase	10	*LOX08pro*	F: TAGTAGCATTGGGCACATA R: TATTAGCGTCTGCGGAGA	[63]
**Autotoxicity and** **saline-alkali** **stresses**	NA	*CmMYB2R40*, *CmMYB2R54*	NA	4	NA	NA	[64]
**Sulphur phytotoxicity**	NA	*MELO3C024245*	Sorting nexin-13	1	NA	NA	[23]

^a^ cleaved amplified polymorphic sequences (CAPSs); not available (NA).

## Data Availability

The original contributions presented in the study are included in the article, and further inquiries can be directed to the corresponding author.

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
