# Peer review of "Molecular Markers for Marker-Assisted Breeding for Biotic and Abiotic Stress in Melon (Cucumis melo L.): A Review"

_ijms, 2024, doi:10.3390/ijms25126307_

Round 1

Reviewer 1 Report

Comments and Suggestions for Authors

The article by Shahwar and co-authors is a good review of molecular markers associated with melon. Since Cucumis melo is an important fruit, the article may be useful in assessing the plant's response to stress factors.

In terms of language, I have no comments. The article was planned and written in a logical manner. The authors first explain the history of Cucumis melo cultivation and the economic importance after which they move on to discuss specific stress factors. 

The authors thoroughly discussed markers related to viruses, bacteria, fungi and nematodes. In contrast, the authors completely omit stress and markers related to attack by insects and other herbivores. This part needs to be supplemented s especially that there are literature data:

https://apsjournals.apsnet.org/doi/10.1094/MPMI-09-11-0252

Author Response

Reviewer 1

The article by Shahwar and co-authors is a good review of molecular markers associated with melon. Since Cucumis melois an important fruit, the article may be useful in assessing the plant's response to stress factors.

In terms of language, I have no comments. The article was planned and written in a logical manner. The authors first explain the history of Cucumis melo cultivation and the economic importance after which they move on to discuss specific stress factors. 

The authors thoroughly discussed markers related to viruses, bacteria, fungi and nematodes. In contrast, the authors completely omit stress and markers related to attack by insects and other herbivores. This part needs to be supplemented s especially that there are literature data:

https://apsjournals.apsnet.org/doi/10.1094/MPMI-09-11-0252

Response: Thank you for your valuable feedback. We appreciate your positive comments on the review of molecular markers associated with melon. Regarding your concern about the omission of stress and markers related to insect and herbivore attacks, we have added this section. We have incorporated literature data, including the reference from https://apsjournals.apsnet.org/doi/10.1094/MPMI-09-11-0252, to ensure a comprehensive discussion of all stress factors.

Reviewer 2 Report

Comments and Suggestions for Authors

The paper is interesting and perfoms a complete relation of all that is known in tomato breeding, including molecular markers. The problem is that the current focus is mainly agronomic. In the recent years there have been a lot of advances related to omics technology, and most of them are being used to breed novel varieties of melon, based on data from metabolomics, or comparing stress sensitive with stress resistant cultivars. I think that authors should include some more data and references related to recent advances in this point and how this would affect melon breeding, to give a more molecular perspective and justify its publications in IJMS rather than Aronomy or Horticulture.

For instance:

https://www.ncbi.nlm.nih.gov/pmc/articles/PMC7143154/

https://pubmed.ncbi.nlm.nih.gov/25892132/

https://pubmed.ncbi.nlm.nih.gov/34804107/

https://pubmed.ncbi.nlm.nih.gov/38612673/

Author Response

Reviewer 2

The paper is interesting and performs a complete relation of all that is known in tomato breeding, including molecular markers. The problem is that the current focus is mainly agronomic. In the recent years there have been a lot of advances related to omics technology, and most of them are being used to breed novel varieties of melon, based on data from metabolomics, or comparing stress sensitive with stress resistant cultivars. I think that authors should include some more data and references related to recent advances in this point and how this would affect melon breeding, to give a more molecular perspective and justify its publications in IJMS rather than Agronomy or Horticulture.

 For instance:

 https://www.ncbi.nlm.nih.gov/pmc/articles/PMC7143154/

https://pubmed.ncbi.nlm.nih.gov/25892132/

https://pubmed.ncbi.nlm.nih.gov/34804107/

https://pubmed.ncbi.nlm.nih.gov/38612673/

Response: Thank you for your thoughtful comments and the references you provided. We appreciate your suggestion to incorporate more data and references related to recent advances in omics technologies and their application in melon breeding. We understand your point regarding the significant role that technologies like metabolomics play in breeding novel varieties of melon, particularly in comparing stress-sensitive and stress-resistant cultivars. However, the focus of this review article is primarily on marker-assisted breeding for biotic and abiotic stress in melon based on molecular markers and molecular breeding rather than metabolomics and proteomics technologies and their application in melon breeding.

While we recognize the importance and relevance of metabolomics and proteomics in the broader context of melon breeding, incorporating these topics extensively may deviate from the core focus of this manuscript. Nevertheless, we have included one of the suggested references, https://pubmed.ncbi.nlm.nih.gov/34804107/ which pertains to drought and salt stress and aligns well with our discussion on stress resistance in melon breeding.

Thank you again for your insights. we hope this clarification addresses your concerns about the scope and focus of our manuscript, ensuring its relevance and fit for publication in IJMS.

Reviewer 3 Report

Comments and Suggestions for Authors

Dear Authors,

The submitted manuscript titled „Molecular markers for marker-assisted breeding for biotic and abiotic stress in melon (Cucumis melo L.): A review” might interest an international audience due to important topic. In my opinion the manuscript is generally well-written, however I have found some imperfections, which (in my opinion) should be corrected or at least  clarified before an eventual publication. I have listed them below:

1.       In my opinion Abstract section should contain references to main parts of the manuscript. Therefore, I suggest to add information about metod of collecting literature sources, as well as at the end the main conclusion.

2.       Lines 32-55. In description of Cucumis melo L. I suggest to add information about pharmacological use of Cucumis melo (see paper of Vishwakarma et al. 2017. Pharmacological importance of Cucumis melo l.: an overview. Asian Journal of Pharmaceutical and Clinical Research 13(3): 8-12.).

3.       I encourage Authors to take into consideration the supplementing chapter Introduction with infomation about current state of knowledge on abiotic and biotic stress of Cucumis melo (see e.g. Wang et al. 2022. Genome-wide characterization of MATE family members in Cucumis melo L. and their expression profiles in response to abiotic and biotic stress. Horticultural Plant Journal, 8(4): 474-488).

4.       Lines 119-130. In my opinion at the end of chapter Introduction the apecific aims of presented review should be listed.

5.       In my opinion the biggest flawn of presented review is lack of chapter Material and methods, presenting the way of collecting literature sources. Were any databases (WoS, Scopus, Google Scholar) used for collecting literature sources? If so, which key words were applied? How the literature sources were included for review (which were the criteria of paper inclusion or exclusion from analysis).

The aforementioned description allows  to avoid an impression that some important literature sources were ommitted in publication.

Author Response

Reviewer 3

Dear Authors,

The submitted manuscript titled “Molecular markers for marker-assisted breeding for biotic and abiotic stress in melon (Cucumis melo L.): A review” might interest an international audience due to important topic. In my opinion the manuscript is generally well-written, however I have found some imperfections, which (in my opinion) should be corrected or at least clarified before an eventual publication. I have listed them below:

  1. In my opinion Abstract section should contain references to main parts of the manuscript. Therefore, I suggest to add information about method of collecting literature sources, as well as at the end the main conclusion.

Response: Thank you for your feedback. To the best of our knowledge, references should not be included in the abstract. Additionally, the journal guidelines specify that the abstract should not contain references. However, we have added detailed information about the method of collecting literature sources as well as the conclusion and future perspective in the manuscript.

  1. Lines 32-55. In description of Cucumis melo L. I suggest to add information about pharmacological use of Cucumis melo (see paper of Vishwakarma et al. 2017. Pharmacological importance of Cucumis melo l.: an overview. Asian Journal of Pharmaceutical and Clinical Research 13(3): 8-12.).

Response: Thank you for your suggestion. We have added information about the pharmacological use of Cucumis melo to the manuscript, including the reference to Vishwakarma et al. (2017), "Pharmacological importance of Cucumis melo.

  1. I encourage Authors to take into consideration the supplementing chapter Introduction with infomation about current state of knowledge on abiotic and biotic stress of Cucumis melo (see e.g. Wang et al. 2022. Genome-wide characterization of MATE family members in Cucumis melo L. and their expression profiles in response to abiotic and biotic stress. Horticultural Plant Journal, 8(4): 474-488).

Response: Thank you for your valuable feedback. We are pleased to inform you that I have already incorporated the reference to Wang et al. (2022) in the manuscript.

  1. Lines 119-130. In my opinion at the end of chapter Introduction the specific aims of presented review should be listed.

Response: Thank you for your suggestion. We have already included the specific aims and objectives of the review in the last paragraph of the Introduction, from lines 122-133.

  1. In my opinion the biggest flawn of presented review is lack of chapter Material and methods, presenting the way of collecting literature sources. Were any databases (WoS, Scopus, Google Scholar) used for collecting literature sources? If so, which key words were applied? How the literature sources were included for review (which were the criteria of paper inclusion or exclusion from analysis).

The aforementioned description allows to avoid an impression that some important literature sources were ommitted in publication.

Response: Thank you for your insightful feedback. We have now added a detailed methodology section after the introduction that includes the names of the databases used (Web of Science, Scopus, Science direct, Research gate, Google Scholar), the specific keywords employed for the search of literature sources. This ensures a comprehensive and transparent approach to literature collection, addressing any concerns about the omission of important sources.

Round 2

Reviewer 3 Report

Comments and Suggestions for Authors

Dear Authors,

In my previous review I suggested to refer in abstract section to main parts of manuscript as aims, methods, main results and main conclusions. I suggest to improve Abstract most important addding information such as e.g. mode of literature selection etc.

Author Response

Response to reviewer’s comment

Reviewer 3

In my previous review I suggested to refer in abstract section to main parts of manuscript as aims, methods, main results and main conclusions. I suggest to improve Abstract most important addding information such as e.g. mode of literature selection etc.

Response: Dear reviewer, I have revised the abstract to clearly delineate the main parts of the manuscript, including the aim, methods, main results, and conclusions. These revisions ensure that the abstract accurately reflects the comprehensive scope and depth of the review and aligns closely with the manuscript's content.